# Ethanolic Extract Propolis-Loaded Niosomes Diminish Phospholipase B1, Biofilm Formation, and Intracellular Replication of *Cryptococcus neoformans* in Macrophages

**DOI:** 10.3390/molecules28176224

**Published:** 2023-08-24

**Authors:** Kritapat Kietrungruang, Sanonthinee Sookkree, Sirikwan Sangboonruang, Natthawat Semakul, Worrapan Poomanee, Kuntida Kitidee, Yingmanee Tragoolpua, Khajornsak Tragoolpua

**Affiliations:** 1Division of Clinical Microbiology, Department of Medical Technology, Faculty of Associated Medical Sciences, Chiang Mai University, Chiang Mai 50200, Thailand; kritapat_k@cmu.ac.th (K.K.); sanonthinee_sookkree@cmu.ac.th (S.S.); sirikwan.sang@cmu.ac.th (S.S.); 2Department of Chemistry, Faculty of Science, Chiang Mai University, Chiang Mai 50200, Thailand; natthawat.semakul@cmu.ac.th; 3Department of Pharmaceutical Sciences, Faculty of Pharmacy, Chiang Mai University, Chiang Mai 50200, Thailand; worrapan.p@cmu.ac.th; 4Center for Research Innovation and Biomedical Informatics, Faculty of Medical Technology, Mahidol University, Salaya, Nakhon Pathom 73170, Thailand; kuntida.kit@mahidol.ac.th; 5Natural Extracts and Innovative Products for Alternative Healthcare Research Group, Faculty of Science, Chiang Mai University, Chiang Mai 50200, Thailand; yingmanee.t@cmu.ac.th; 6Department of Biology, Faculty of Science, Chiang Mai University, Chiang Mai 50200, Thailand

**Keywords:** propolis, niosomes, pulmonary cryptococcosis, *Cryptococcus neoformans*, phospholipase B1, biofilm formation, phagocytosis

## Abstract

Secretory phospholipase B1 (PLB1) and biofilms act as microbial virulence factors and play an important role in pulmonary cryptococcosis. This study aims to formulate the ethanolic extract of propolis-loaded niosomes (Nio-EEP) and evaluate the biological activities occurring during PLB1 production and biofilm formation of *Cryptococcus neoformans*. Some physicochemical characterizations of niosomes include a mean diameter of 270 nm in a spherical shape, a zeta-potential of −10.54 ± 1.37 mV, and 88.13 ± 0.01% entrapment efficiency. Nio-EEP can release EEP in a sustained manner and retains consistent physicochemical properties for a month. Nio-EEP has the capability to permeate the cellular membranes of *C. neoformans*, causing a significant decrease in the mRNA expression level of *PLB1*. Interestingly, biofilm formation, biofilm thickness, and the expression level of biofilm-related genes (*UGD1* and *UXS1*) were also significantly reduced. Pre-treating with Nio-EEP prior to yeast infection reduced the intracellular replication of *C. neoformans* in alveolar macrophages by 47%. In conclusion, Nio-EEP mediates as an anti-virulence agent to inhibit PLB1 and biofilm production for preventing fungal colonization on lung epithelial cells and also decreases the intracellular replication of phagocytosed cryptococci. This nano-based EEP delivery might be a potential therapeutic strategy in the prophylaxis and treatment of pulmonary cryptococcosis in the future.

## 1. Introduction

Pulmonary cryptococcosis is an opportunistic and invasive mycosis usually found in immunocompromised patients [1]. The pathogenesis usually originates from *Cryptococcus neoformans* through inhaling spores and small infective particles, ultimately resulting in respiratory infection [2]. Several virulence factors, such as polysaccharide capsules and degrading enzymes, are produced to allow yeast pathogen adhesion, invasion, and damage to the host cells. Among the virulence-associated enzymes, secretory phospholipase B1 (PLB1) plays a crucial role in facilitating the adhesion and destabilization of the host cell membrane and the phospholipid lung surfactant [3,4]. PLB1 is also associated with the escape of yeast cells from the pulmonary macrophages through nonlytic exocytosis or vomocytosis [5,6]. Additionally, the extracellular polymeric matrix (EPM), or biofilm, constitutes the dynamic communities of microorganisms via adhesion/matrix proteins signaling and directional proliferation of the original-adhered yeast cells [7]. The biofilm formation begins with yeast cell adhesion, releasing enzyme, and biofilm maturation [8]. It promotes the survival of yeast cells and protects them from host immunity as well as antifungal drugs [9]. Although the treatment of pulmonary cryptococcosis is currently based on the first-line antifungal drug amphotericin B (AMB), the adverse effects, especially nephrotoxicity, remain a serious concern [10]. Hence, the application of a natural antimicrobial agent has become increasingly attractive as an alternative treatment.

Propolis is a complex mixture of natural and resinous substances that contain various active ingredients such as gallic acid, quercetin, pinocembrin, chrysin, and galangin [11]. Notably, it is a rich source of therapeutic properties and has antioxidant, anti-inflammatory, antimicrobial, and immunomodulatory capabilities [12,13,14]. Several reports have revealed potent in vitro anti-fungal activity of the ethanolic extract propolis, or EEP, against *Candida albicans* [15] and *C. neoformans* [16]. Moreover, our previous work found the anti-cryptococcal activity of EEP to reduce the growth rate and major virulence factors of *C. neoformans*, including the polysaccharide capsule, melanin, and urease [17]. Subsequently, the encapsulation of EEP based on the poly (n-butyl cyanoacrylate) (PBCA) nanosystem was further explored for therapeutic application in cryptococcal meningoencephalitis through blood–brain barrier (BBB) drug delivery [18]. However, administration via the pulmonary route is restricted due to mucus and lung surfactant [19]. To achieve therapeutic use in the lungs, non-ionic surfactant vesicles, or niosomes, were considered as an effective nanocarrier of EEP.

Niosomes are composed of self-assembled non-ionic surfactants and cholesterol, containing a hydrophilic head and hydrophobic tail, to form a vesicle. The niosomes are modified with a liposomal vesicle usually made of phospholipids. Due to the fragile phospholipid membranes, the liposome has low physical stability compared to the niosome and can cause drug leakage [20]. Moreover, the niosomal structure contains lipids, a similar component of lung surfactant, which improves nanoparticle penetration in the mucus layer and provides sustainable drug release [21]. These advantages of niosomes are beneficial to the encapsulation of both hydrophobic and lipophilic molecules [22]. Previously, the formulation of AMB–niosomes was established by significantly reducing the fungal burden, in an animal model, with invasive pulmonary aspergillosis [23]. In addition, the encapsulation of EEP was also successfully achieved in niosomes against *Staphylococcus aureus*, *C. albicans* [18], and *Mycobacterium tuberculosis* [24,25]. Therefore, this study aims to investigate the efficacy of EEP through the niosome system against *C. neoformans* for an in vitro model of pulmonary cryptococcosis.

In this work, we fabricated Nio-EEP and evaluated its anti-virulence factors, including PLB1, and the biofilm formation in vitro. Furthermore, the influence of Nio-EEP-induced phagocytosis and the killing of *C. neoformans* by macrophages were also investigated.

## 2. Results

### 2.1. Physicochemical Characterization of Nio-EEP

#### 2.1.1. Particle Size, Polydispersity Index (PDI), Zeta Potential (ZP), Entrapment Efficiency (EE), Loading Capacity (LC), and Morphology

Niosomes were successfully fabricated with different proportions of non-ionic surfactant and CHOL. As shown in Table 1, all niosomal formulations had an average particle size of approximately 100–280 nm with PDI 0.32–0.37. The ZP measurements exhibited a negative surface charge of niosomal formulations, ranging from −10 to −12 mV. The EE of each formulation was greater than 85% while the LC showed a variation. The highest LC was approximately 83%, observed in the F1 formulation; in contrast, F2 and F3 had a lower capacity of approximately 46–50%. Due to the highest LC, F1 was chosen as the suitable formulation to characterize the physicochemical structure further. The particle number of F1 was calculated according to the previous study [26]. The number of particles was 6.5 × 10^11^ vesicles/mL corresponding to 3.52 ± 0.01 mg/mL of EEP. The scanning transmission electron microscope (STEM) images displayed the morphological particles in nanometer scales and presented the spherical vesicles as shown in Figure 1a,b.

#### 2.1.2. Chemical Composition

To verify the existence of EEP in niosomes, nuclear magnetic resonance (NMR) spectroscopy might be a reliable tool, especially for studying the encapsulation of EEP into niosomal vesicles. The NMR spectra of Tween 80 (TW80), Span 60 (SP60), cholesterol (CHOL), and niosomes were recorded using DMSO-*d*_6_ as a solvent, as shown in Figure 1c. The NMR spectra of TW80 and SP60 showed similar chemical shifts around 1–4 ppm corresponding to aliphatic protons [27]. It should be noted that TW80 featured chemical shifts around 4–5 ppm which ascribes to the olefinic protons [28]. The NMR spectrum of CHOL showed chemical shifts in two regions, i.e., chemical shifts around 1–2.5 ppm which are ascribed to aliphatic protons and chemical shifts around 4–5.5 ppm which are ascribed to olefinic protons [29]. Nio also showed a profile of absorption peaks similar to the surfactants and CHOL, indicating the successful formulation of niosomes. Next, the NMR analysis of Nio, EEP, and Nio-EEP was individually recorded, as shown in Figure 1d. The ^1^H NMR spectrum of EEP revealed all phytochemical and other chemical constituents that can be interpreted based on chemical shift fingerprints. Examples of molecules with chemical shifts of aliphatic protons in the 0.5–3.0 ppm range include terpenoids, steroids, and linear fatty acid side chains for fats, oils, and waxes. Additionally, peaks in the chemical shift range of 3.5–5.5 ppm are due to sugar components. It is important to note that HPLC typically does not detect these compounds, thereby rendering NMR a useful alternative. Interestingly, chemical shifts around 6.0–8.1 ppm are also observed which correspond to the protons belonging to aromatic phenolic compounds [30,31]. It was found that the ^1^H NMR spectrum of the Nio-EEP sample displayed chemical shifts that resembled the niosome components as well as the EEP components at a chemical shift between 7.25 and 7.75 ppm. In the HPLC analysis, our EEP sample consists of several phenolic compounds [11]. This result implies that phenolic compounds were successfully encapsulated into the niosomal formulation.

#### 2.1.3. In Vitro Release Study

The study of EEP released from niosomal vesicles was carried out in modified stimulated lung fluid (mSLF) at pH 6.6 which mimics the pathological conditions of pneumonia. As shown in Figure 1e, the initial burst release of EEP was 11.2% in the first 3 h followed by a gradual release for up to 24 h, indicating that Nio-EEP acts as a sustained-release formulation. According to these physicochemical characteristics, F1 provided preferable properties for further investigation of its bioactivity against *C. neoformans*.

#### 2.1.4. Stability Testing

The stability of the formulations was determined after storage at 4 °C for 1 month. As shown in Figure 2, the average size, PDI, and ZP values showed slight alterations at different times. Notably, EEP remained the same in the formulation; approximately 88% of EE was not different from the initial time point. This result indicates that the Nio-EEP and Nio were stable for 1 month of storage at 4 °C.

### 2.2. In Vitro Biological Activity of Nio-EEP

#### 2.2.1. Cytotoxicity Assay of Niosomes

The cytotoxicity of Nio-EEP was evaluated on A549 and NR8383 cells after treatment with nanoparticles between 0.325 and 6.5 × 10^11^ vesicles/mL. The results show a significant reduction in metabolic activity in both cell lines when treated with nanoparticles between 3.25 and 6.5 × 10^11^ vesicles/mL (Figure 3a,b). Niosomes with a number of particles below 3.25 × 10^11^ vesicles/mL are considered non-cytotoxic. Therefore, these concentrations of Nio-EEP were selected for investigation in further experiments.

#### 2.2.2. Anti-Fungal Susceptibility Testing

The inhibitory effects of Nio-EEP on the growth of yeast cells was evaluated by a colony forming unit (CFU) assay. The results established that none of the concentrations of Nio-EEP and Nio reduced the growth of *C. neoformans* (Figure 3c). On the other hand, the metabolic activity was reduced based on a 3-[4,5-dimethylthiazol-2-yl]-2,5 diphenyltetrazolium bromide (MTT) assay. As shown in Figure 3d, there was no statistically significant difference in the metabolic activity of yeast cells between the control and Nio groups. Remarkably, Nio-EEP significantly reduced the metabolic activity of the yeast cells by approximately 25% and 40% at 1.0 and 2.0 × 10^11^ vesicles/mL, respectively, in contrast to Nio. Based on the results, Nio-EEP has the efficacy to inhibit the metabolic activity of *C. neoformans*.

#### 2.2.3. Localization of Nio-EEP 

To ensure the uptake of niosomes by the yeast cells, an assessment of niosome localization was performed. As shown in Figure 4a, the Nio-EEP was tracked by Nile red (NR) labeling (red) while the yeast cells were stained with calcofluor white (CFW) (blue). After the incubation period, the accumulation of Nio-EEP was observed inside the yeast cytoplasm. The orthogonal imaging analysis confirmed that Nio-EEP was located within the yeast cells (Figure 4b) and therefore could be up-taken by the yeast cells.

### 2.3. Anti-Virulence Factors of Nio-EEP

#### 2.3.1. Phospholipase Production

Enzymatic phospholipase activity has been found to promote the binding of *C. neoformans* during lung infection; therefore, the effect of Nio-EEP on yeast phospholipase activity was preliminarily assessed. It was found that the phenotypic phospholipase activity of yeast was not reduced by niosomes, as determined by the EYA assay (Figure 5a). While the genotypic determination of the phospholipase-related gene, *PLB1*, showed a significant reduction in expression level after treatment with niosomes, both particle concentrations of Nio did not significantly affect *PLB1* expression. Interestingly, at 2 × 10^11^ vesicles/mL, Nio-EEP exhibited a significant down-regulation of *PLB1* levels by 0.54-fold changes in contrast to Nio (Figure 5b). These results imply that EEP might contribute to the interference of phospholipase production at the transcriptional level, leading to an attenuation of the virulence factor.

#### 2.3.2. Biofilm Formation

Following adhesion on lung epithelial cells, a biofilm of *C. neoformans* is formed and consequently self-produces an extracellular polymeric matrix (EPM) as a defense mechanism. To investigate the effects of Nio-EEP on biofilm formation, an examination of the formation of biofilm was conducted by MTT assay and confocal laser scanning microscope (CLSM) imaging analysis. There was no statistically significant difference in the biofilm formation of yeast cells between the control and Nio groups. Interestingly, the production of yeast biofilm was significantly reduced by 54 and 57% after treatment with 1.0 and 2.0 × 10^11^ vesicles/mL of Nio-EEP, respectively, in contrast to Nio (Figure 6a). To ensure that the formed biofilm was diminished by Nio-EEP, the three-dimensional (3D) structure of biofilm was evaluated by CLSM imaging analysis. The fluorescent images presented the metabolically active yeast cells with FUN-1 (red) and EPM with Concanavalin A (Con A) Alexa Flour 488 conjugate (green). The biofilm thickness of the yeast control, Nio, and Nio-EEP was approximately 25, 23, and 16 µm, respectively. The results exhibit, interestingly, that the biofilm thickness of Nio-EEP-treated yeasts was clearly reduced by about 30% in contrast to Nio, as shown in Figure 6b. These findings suggest that Nio-EEP might interrupt the mitochondrial activity and lead to a decrease in biofilm formation.

Aside from the phenotypic change in biofilm formation, the molecular expression levels of biofilm-related genes, including the *UGD1*, *UXS1*, and *MAN1* genes, were further assessed. As shown in Figure 6c, Nio did not statistically affect the expression of these three genes compared with the yeast control. Nio-EEP remarkably suppressed the mRNA expression levels of *UGD1* and *UXS1* but did not change the *MAN1* gene expression. The yeast cells treated with 1.0 × 10^11^ and 2.0 × 10^11^ vesicles/mL of Nio-EEP showed a significant down-regulation of mRNA levels by approximately 0.35 and 0.24-fold changes for *UGD1* and 0.86 and 0.40-fold changes for *UXS1*, respectively, contrasting the results with Nio. These findings suggest that Nio-EEP might influence the production of biofilm through the down-regulation of the *UGD1* and *UXS1* genes.

#### 2.3.3. Nio-EEP-Induced Intracellular Killing

To evaluate the function of macrophages to phagocytose the treated yeasts, a phagocytosis assay was conducted. The NR8383 cells were challenged with the Nio-EEP-treated yeasts and stained with Wright-Giemsa. The phagocytosed yeast cells were presented (Figure 7a) and the percentages of phagocytosis were approximately 27–41% in all groups (Figure 7b) while the phagocytosis index was 0.3–0.4 cells/macrophage, as shown in Figure 7c. These results indicate that treatment with Nio-EEP and Nio did not affect the phagocytosis activity of macrophages. From these findings, we then hypothesized whether treating *C. neoformans* with Nio-EEP would decrease the survival rate of yeast cells in macrophages and the survival of intracellular yeasts was then carried out by a CFU assay. As shown in Figure 7d, the survival rate of Nio-EEP-treated yeasts was reduced by 20% compared to Nio. Noticeably, the survival rate of Nio-EEP-treated yeasts was reduced by 47% when compared to the yeast control. Based on the results, Nio-EEP could induce the killing of intracellular *C. neoformans* by alveolar macrophages.

## 3. Discussion

The World Health Organization (WHO) has reported *C. neoformans* as one of the most critical fungal pathogens causing the greatest threat to human health [32]. Pulmonary cryptococcosis remains a significant concern, especially in an immunocompromised patient. Management of this invasive cryptococcal infection currently relies on the first-line drug, AMB. However, AMB-induced nephrotoxicity is a principal issue that limits effective treatment [1]. Thus, alternative treatments derived from natural substances have gained more attention as therapeutic agents. EEP, a natural product from bees, is a source of several effective molecules that exhibit antimicrobial activity [11]. Our previous study reported that EEP was found to have anti-fungal properties against major cryptococcal virulence factors, such as polysaccharide capsules, melanin pigment, and urease [17]. However, the direct application in the pulmonary system is still restricted due to the water solubility of the EEP [33] and lung surfactant permeability [34].

Improving the therapeutic properties of EEP was achieved through a nanocarrier-based drug delivery system. Niosomes are lipid-based nanocarriers produced from non-ionic surfactants and CHOL, which might be useful for delivery to the pulmonary system. This study investigated niosomal nanocarriers using different ratios of non-ionic surfactants (SP60 and TW80) and the additive CHOL. The non-ionic surfactant structure with a single alkyl tail generally forms a niosomal vesicle in aqueous solutions [28] and the insertion of CHOL into the niosomal membrane requires the membrane rigidity to increase, stabilizing the vesicular structure [35,36,37]. It was reported that these non-ionic surfactants were greater for high encapsulation of natural products [38,39] and that the EEP was successfully encapsulated by SP60 or TW80 with approximately 70% EE [24,40]. In this study, the optimization of the different SP60 and TW80 concentrations was initiated according to Sangboonruang et al. [41]. Moreover, the ratio of surfactants was further optimized by reducing the concentrations of both SP60 and TW80 (F1) as well as SP60 (F3) only. In the case of reducing only TW80, we found the undesirable characteristic [41].

As a result, the niosomal dispersion exhibited vesicular particle sizes of Nio ranging from 108 to 255 nm while Nio-EEP was 152 to 268 nm. The mean particle size of Nio did not change between the formulations F1 and F3. This finding means that a double concentration of TW80 in F3 did not affect the particle size. However, the particle size of F2 was reduced due to high surfactant concentrations, possibly inducing micelle, rather than vesicle formations [42,43]. In addition, Nio-EEP particle size was markedly larger than Nio in F1 and F2. This might be due to EEP being encapsulated into the hydrophobic layer, leading to increased particle size [41]. Conversely, F3 exhibited a mean particle size of Nio-EEP smaller than Nio, likely due to the interaction between the surfactant and the extract, enhancing niosomal cohesion and resulting in a decreased vesicle size [44,45].

The distribution of vesicles in the solution was determined with PDI values by DLS analysis. The PDI values of all three formulations ranged from 0.32 to 0.37, indicating a relatively homogeneous vesicle population [41]. Furthermore, the presence of a surface charge, or ZP, can produce a repulsive force between the vesicles, causing a distributed suspension. In theory, ZP values outside of −30 mV to +30 mV are generally considered to have sufficient repulsive force, attaining better physical colloidal stability [46]. However, our results showed the ZP values of all formulations with a negative charge, ranging from −9.38 to −10.54 mV. To further improve the ZP values, some modifications with charge-inducing agents, such as diacetyl phosphate (DCP) for negative charges or stearyl amine (STR) for positive charges, [47] may be needed.

Regarding EE, our formulations of TW80, SP60, and CHOL exhibited high encapsulation rates of more than 85% in all formulations. In other works, it was reported that similar components produced from TW80 and CHOL were at 70% EE [40] and from SP60 and CHOL were at 71.29% EE [24]. Differences in the efficiency of drug encapsulation may depend on several factors, such as 3D chemical structure, hydrophilicity, the ratio of surfactant, and the structure of the surfactant [48,49]. In addition, the CHOL distributed between the lipid bilayer enhances encapsulation due to its membrane-stabilizing effect and the prevention of drug leakage [47,50]. The capacity of the nanovesicle to load EEP is determined by %LC. F2 and F3 contained LC below 50%, even after increasing the lipid phase composition, indicating the maximum capacity of the EEP–lipid interaction. The appropriate niosomal composition of F1 could occupy the EEP and result in the highest % LC. Therefore, F1 was chosen for further investigation.

The physicochemical characteristics of niosomes were confirmed by STEM and NMR spectroscopy, as shown in Figure 2. The F1 formulation showed spherical morphology with the approximate particle size correlating to the DLS results. In addition, the NMR spectra of Nio, EEP, and Nio-EEP exhibited minor constituents of EEP at a region between 7.25 and 7.75 ppm, corresponding to EEP derived from phenolic compound regions, as reported by Ilhan-Ayisigi et al. [40]. Hence, these results support the encapsulation of EEP in the niosomal system. The stability of the obtained formulation was also tested under storage conditions of 4 °C for 1 month. The particles’ size, PDI, and ZP did not change and the loaded EEP was retained in the nanovesicles by more than 85% throughout the study period. This indicates a stable property of this nanoformulation. A drug release profile is one of the most important characteristics describing the process of payload migration from the niosome to the outer system [51]. In this study, the in vitro release profile of Nio-EEP revealed an initial burst-release in the mSLF at 3 h and a sustained release during the experiment period of 24 h. Additionally, the in vitro toxicity of the niosomes was not found in the A549 and NR8383 cells at a number of particles below 3.25 × 10^11^ vesicles/mL. The overloaded number of NPs with induced cellular cytotoxicity had been described elsewhere [52].

For anti-fungal activity, Nio-EEP vesicles did not affect the growth of *C. neoformans* in terms of the colony count. However, Nio-EEP exhibited an inhibitory effect on the metabolic activity of *C. neoformans* at approximately 40%. Based on the MTT assay reflecting the metabolic activity rather than direct cell viability, we suggest that Nio-EEP had the anti-fungal ability through the interference of yeast mitochondrial function. In concordance with previous reports, the released EEP from Nio-EEP might potentially interfere with the mitochondrial enzyme activity [11,53] and inhibit the electron transport chain (ETC) complexes (complex I to V), eventually resulting in mitochondrial dysfunction [54]. The Nio-EEP intracellular uptake in the yeast cells was visualized by CLSM. According to a previous study, this evidence can be described by non-phagocytic eukaryotic cells having an uptake nanoparticle size ranging from 200–500 nm via endocytosis [55]. One of the endocytosis processes in the yeasts might be clathrin-mediated endocytosis, initiated by cytosolic proteins assembling to promote plasma membrane blending and transforming the flat plasma membrane to clathrin-coated vesicles [56]. Consequently, it might be implied that the reduction of metabolic activity is capable of releasing intracellular EEP.

Nio-EEP presented properties against important virulence factors related to the adhesion and biofilm production of *C. neoformans* and PLB1 is one of the virulence-associated enzymes that play a crucial role in promoting yeast cell adhesion on the pulmonary epithelial cell surface [57]. Using a screening method with an EYA assay, the PLB1 production of niosome-treated *C. neoformans* was not found. The EYA method is based on precipitation zone production which has a low sensitivity [58,59]. Even though the radiolabeling method is specific to detect PLB1 activity, there are more practical difficulties. Thus, gene expression analysis is recommended and considered to be an accurate evaluation method [57,58]. As a result, *PLB1* expression at the transcriptional level was significantly disrupted by Nio-EEP. Therefore, the regulation of phospholipase synthesis might be defective, and eventually, the PLB1 enzyme activity was reduced. 

Once the yeast cells adhere to and colonize the epithelial host cells, cell-to-cell communication can lead to the formation of EPM or biofilm. Yeast biofilm is another virulence factor that provides defensive activity from anti-fungal drug penetration and the pulmonary immune response, thus promoting yeast survival. In this work, the biofilm production indicating the yeast community was significantly decreased by approximately 50%. Also, the physical structure of yeast biofilm was reduced from 25 μm to 16 μm in thickness while Nio did not have this effect. These observations could imply that the biofilm decreased as a result of releasing EEP. In agreement with previous work by Kumari et al., it was found that a phenolic compound had reduced the biofilm formation of *C. neoformans*. It can be explained that the phenolic compounds trigger reactive oxygen species (ROS) generation and oxidative stress, sequentially reducing EPM biosynthesis [60]. Likewise, Iadnut et al. reported that the biofilm mass and gene-related expression of biofilms in *C. albicans* were reduced by the EEP-loaded PLGA-NPs [11]. Major components related to EPM in the biofilm are glycosyl compositions, such as mannose, glucuronic acid, and xylose, and these sugar molecules are processed through glycan synthetic pathways. The nucleotide sugars are the donor molecules for structural polysaccharide capsule synthesis. Guanosine diphosphate-mannose (GDP-Man) is made through the sequential action of phosphomannose isomerase (MAN1), which is encoded by the *MAN1* gene [61]. Uridine diphosphate-glucuronic acid (UDP-GlcA) is produced through the dehydrogenase uridine diphosphate-glucose (UDP-Glc) pathway by uridine diphosphate-glucose dehydrogenase (UGD1). UDP-GlcA is sequentially decarboxylated by uridine diphosphate-xylose decarboxylase (UXS1) to produce the uridine diphosphate-xylose (UDP-Xyl) [62,63]. For a clearer understanding, the mRNA expression of genes associated with the formation of *C. neoformans* biofilm was further assessed. The expression of *UGD1* and *UXS1* mRNA was suppressed in Nio-EEP-treated biofilm while *MAN1* had no change. These findings can be explained by the fact that the phenolic compounds of the released EEP may be interacting with glycosyltransferase, leading to the lack of a specific sugar donor to supply the downstream product of the EPM biosynthesis [60,64,65]. This suggests that Nio-EEP has the ability to disrupt the yeast community structure and inhibit the formation of biofilm via the glycosyl component interruption. To fill this gap, the proteomic profiles in mature biofilm, metabolic product accumulation, and a quorum sensing mechanism should be further studied. The integration of multidisciplinary fields will promote more understanding and development of strategies to reduce the virulence of fungal pathogenesis and prevent the extrapulmonary dissemination of *C. neoformans*.

Additionally, Nio-EEP was further investigated for its ability to enhance phagocytosis or kill yeast via alveolar macrophages. The results demonstrated no differences in the phagocytosis rate and phagocytosis index, suggesting Nio-EEP did not affect yeast recognition and phagocytosis via Fc–FcγR interactions [66]. In consideration of the survival of phagocytosed yeast cells, we found that Nio-EEP significantly decreased the survival rate of *C. neoformans*. We also suspect that the lower growth and survival rate of intracellular *C. neoformans* is due to the decrease in PLB1 activity. Taken together, the reduction of urease and melanin induced by EEP has been reported [17,67]. *C. neoformans* inhibited the acidification of the phagolysosome by the urease enzyme which degrades urea into CO_2_ and ammonia. Melanin also plays a protective role against free radicals [68]. Additionally, phospholipase serves as a phospholipid hydrolysis enzyme in the phagolysosome [69]. These virulence factors are involved in cryptococcal survival, yeast escape, and macrophage killing. Therefore, the reduction of virulence factors not only detriments yeast survival but also increases sensitivity to killing via hydrolytic enzymes, reactive oxygen species, and reactive nitrogen species (ROS/RNS). To fill this knowledge gap, the investigation of PLB1 activity and other virulence factors, as well as the host immune response to the intracellular pathogen with Nio-EEP, should be further performed.

## 4. Materials and Methods

### 4.1. Materials 

Propolis powder was kindly provided by Bee Product Industry Co., Ltd., Lamphun, Thailand. Sorbitan monostearate (Span 60; SP60), polysorbate 80 (Tween 80; TW80), and cholesterol (CHOL) were purchased from Sigma-Aldrich (St. Louis, MO, USA). All other chemicals and reagents used in this study were of analytical grade, including Sabouraud Dextrose Agar (SDA) (HiMedia, Mumbai, India), Sabouraud Dextrose Broth (SDB) (HiMedia, Mumbai, India), Dulbecco’s modified Eagle’s medium (DMEM) (Gibco, Carlsbad, CA, USA), Kaighn’s modification of Ham’s F12 medium (F-12K) (Caisson laboratories Inc., Smithfield, UT, USA), Chloroform (RCI Labscan, Taipei, Taiwan), Potassium phosphotungstic acid (TED PELLA Inc., Redding, CA, USA), 3-[4,5-dimethylthiazol-2-yl]-2,5 diphenyltetrazolium bromide (MTT) (Bio Basic Inc., Markham, ON, Canada), Calcofluor white (CFW) (Sigma-Aldrich, St. Louis, MO, USA), Nile-red dye (Sigma-Aldrich, St. Louis, MO, USA), ProLong Gold anti-fade reagent (Thermo Fisher Scientific, Waltham, MA, USA), Egg Yolk Tellurite Emulsion (HiMedia, Mumbai, India), FUN-1 (Molecular Probe, Waltham, MA, USA), Concanavalin A (Con A)-Alexa Flour 488 conjugate (Thermo Fisher Scientific, CA, USA), lipopolysaccharide (LPS) (Sigma-Aldrich, MO, USA), and interferon-γ (IFN-γ) (Biolegend, San Diego, CA, USA).

### 4.2. Yeast and Cell Lines

*C. neoformans* H99 was kindly provided by Assoc. Prof. Pojana Sriburee (Department of Microbiology, Faculty of Medicine, Chiang Mai University, Chiang Mai, Thailand). Yeast cells were maintained on SDA and incubated at 37 °C for 72 h. A few isolated colonies were selected, cultured in SDB, incubated at 37 °C for 16–18 h, and then shaken before experimentation.

The human lung epithelial cancer cell line (A549) was kindly obtained from Asst. Prof. Dr. Khanittha Punturee (Department of Medical Technology, Faculty of Associated Medical Sciences, Chiang Mai University, Chiang Mai, Thailand) and was cultured in DMEM supplemented with 10% (*v*/*v*) fetal bovine serum (FBS), 100 units/mL of penicillin, and 100 µg/mL of streptomycin. Alveolar macrophage cell line (NR8383) (ATCC, Manassas, VA, USA) was cultured in F-12K supplemented with 15% (*v*/*v*) FBS, 100 units/mL of penicillin, and 100 µg/mL of streptomycin. The cells were maintained in a humidified atmosphere of 5% CO_2_ at 37 °C.

### 4.3. Formulation of Nio-EEP

Niosomal formulations were prepared by the thin–film hydration (TFH) technique according to our previous study with some modifications [41]. Briefly, different molar ratios of SP60, TW80, and CHOL were dissolved in 9 mL chloroform and supplemented with 1 mL ethanol solution of EEP (20 mg/mL) for Nio-EEP or without EEP for empty niosomes (Nio) in a round-bottom flask. The organic solvent was evaporated using a rotary evaporator under a vacuum at 60 °C and 100 rpm rotation to obtain a thin lipid film on the inner flask wall. The lipid thin film was hydrated with 10 mL PBS, pH 7.4, under mechanical stirring at 60 °C for 30 min. The obtained niosomal suspension was then subjected to an ultrasonic probe sonicator (Hielscher UP50H, Wanaque, NJ, USA) at 80% amplitude for 30 min in an ice bath to achieve size reduction. The constituents of the different niosomal formulations are indicated in Table 2. A schematic representation of a Nio-EEP is shown in Figure 8.

### 4.4. Physicochemical Characterization of Niosomes

#### 4.4.1. Particle Size, Polydispersity Index (PDI), Zeta Potential (ZP), Morphological Analysis, and Stability Testing

The particle size, PDI, and ZP of niosomes were measured using a Malvern Zetasizer Nano ZSP system (Malvern Instruments, Worcestershire, UK). Niosomal samples were suspended in PBS at a 1:100 dilution. The analyses were performed based on triplicates in three individual runs. 

The morphological characteristics of niosomal vesicles were examined by scanning transmission electron microscope (STEM). A drop of the niosomal sample was placed onto a carbon-coated copper grid and stained with 1% (*w*/*v*) phosphotungstic potassium acid aqueous solution. The morphology was observed by JSM-IT800 Ultrahigh Resolution Field Emission SEM (JEOL, Peabody, MA, USA).

To investigate the stability of the formulations, the niosomal vesicles were kept at 4 °C, and measured in size, PDI, and ZP on days 7, 14, and 30. 

#### 4.4.2. Nio-EEP Chemical Structure

To further verify that the Nio-EEP was successfully performed, the chemical structure of niosomal vesicles was investigated. Nio-EEP, Nio, and EEP were lyophilized before the analysis comparison to the samples with chemical standards. All samples were dissolved in 700 µL deuterated dimethyl sulfoxide (DMSO-*d*_6_) then filtered into a nuclear magnetic resonance spectroscopy (NMR) tube. ^1^H NMR spectra were recorded by 500 MHz NMR spectroscopy (Bruker AV-500 NEO^TM^, Berlin, Germany) and are internally referenced to residual proton signals in DMSO-*d*_6_ (2.50 ppm).

#### 4.4.3. Entrapment Efficiency (EE) and Loading Capacity (LC) 

Free EEP was determined after separation from Nio-EEP by the centrifugation method with a membrane molecular weight cut-off (MWCO) filter, 10 kDa at 8000× *g*, and 4 °C for 2 h 30 min [41]. Then, the niosomal residues were re-suspended in 1 mL of sterile PBS, pH 7.4. Finally, the un-entrapped filtrate solution was determined at 290 nm by a UV/Vis spectrophotometer (Specord Plus, Jena, Germany) and the EEP concentration was calculated using the EEP standard calibration curve.

The percentages of entrapment efficiency (%EE) and loading capacity (%LC) were calculated by the following equation [70]:(1)$$\%\mathrm{EE}=\left[\frac{\left(C_{t}-C_{f}\right)}{C_{t}}\right]\times \;100$$
where C_t_ is the concentration of total EEP and C_f_ is the concentration of free EEP in filtrate.

The amount of EEP-loaded per weight unit of lipid phase was calculated as shown on the %LC [71]:(2)$$\%\mathrm{LC}=\frac{A_{t}}{L_{t}}\times \;100$$
where A_t_ is the total amount of Nio-EEP and L_t_ is the total weight of lipid phase.

EEP retained in the formulation corresponding to % EE was determined at 4 °C on days 7, 14, and 30.

#### 4.4.4. In Vitro Release Study 

The release of EEP from the niosomes was investigated using the modified dissolution method [11]. Briefly, the Nio-EEP was dissolved in 3 mL modified and stimulated lung fluid (mSLF) solution [72] and adjusted to pH 6.6 to mimic acidic pathological conditions [73]. The samples were rotated at 37 °C and the supernatant was collected by centrifugation at different time points and then replaced with the same volume of fresh mSLF solution. The amount of released EEP was analyzed using a UV/Vis spectrophotometer at 290 nm and compared with the EEP standard calibration curve.

### 4.5. In Vitro Bioactivity of Nio-EEP

#### 4.5.1. Cytotoxicity Assay

The cytotoxicity of the niosomal formulation was examined on A549 and NR8383 cells by MTT assay [74]. Briefly, A549 (1 × 10^4^ cells/well) or NR8383 (1 × 10^5^ cells/well) was seeded in a 96-well tissue culture plate and cultured for 24 h. In addition, NR8383 (1 × 10^5^ cells/well) was cultured in a 96-well tissue culture plate for 48 h. Then, various numbers of nanoparticles were added to the cells. After another 24 h incubation, 20 μL of MTT solution (5 mg/mL) were added to the treated cells and incubated at 37 °C for 4 h. Then, the supernatant was removed and 200 μL of dimethyl sulfoxide (DMSO) were added to solubilize the MTT-formazan produced by living cells. The optical density (OD) was measured at 540 and 630 nm. The metabolic activity was further evaluated and calculated by the equation shown below.
(3)$$\%\mathrm{Metabolic}\;\mathrm{activity}=\frac{{(\mathrm{OD}}_{540}-{\mathrm{OD}}_{630})}{\left({\mathrm{OD}}_{540}-{\mathrm{OD}}_{630}\right)}\;\times \;100$$

#### 4.5.2. Anti-Fungal Susceptibility Testing

The antifungal activity of Nio-EEP was determined by modified the. Clinical Laboratory Standard Institute (CLSI) broth microdilution method (M27-A3) [17]. In brief, the Nio-EEP (or Nio) was serially diluted with various numbers of particles in a 96-well microtiter plate. Then, 100 µL of yeast suspension (1 × 10^3^ CFU/mL) were seeded and incubated at 37 °C for 72 h. After incubation, the yeast proliferation was determined by a colony forming unit (CFU) counted on an SDA plate and the metabolic activity was further examined by a MTT assay as previously described.

#### 4.5.3. Yeast Cell Uptake of Nio-EEP 

Nio-EEP uptake by yeast cells was examined using a confocal laser scanning electron microscope (CLSM). The yeast cells (1 × 10^8^ CFU/mL) were stained with 2 mg/mL CFW and incubated for 30 min in the dark at room temperature. Then, the CFW-labeled yeast cells were washed and re-suspended in PBS. Meanwhile, 250 μL of Nio-EEP were stained with 40 μL of Nile Red (NR) solution (0.25 mg/mL) in PBS and incubated for 30 min in the dark at room temperature. Following washing with PBS, the NR-labeled Nio-EEP was re-suspended in PBS and further incubated with the CFW-labeled yeast cells. After 3 h incubation at 37 °C, the excess niosomes were removed, re-suspended with ProLong Gold anti-fade reagent and analyzed by CLSM (LSM900 Airyscan 2; Zeiss, Oberkochen, Germany).

### 4.6. Effect of Nio-EEP on Virulence Factors of C. neoformans

#### 4.6.1. Phospholipase Production 

The phenotypic phospholipase enzyme activity was examined using the egg yolk agar (EYA) method [75]. The yeast cell suspension (1 × 10^8^ CFU/mL) was pre-treated with Nio-EEP (or Nio) and incubated at 37 °C for 4 h with rotation. Following washing with PBS, the concentration of treated yeast cells was adjusted. Five microliters (1 × 10^6^ cells) were dropped on EYA and incubated at 37 °C for 4 days. The precipitation zone and the diameter of the colony was measured and the phospholipase production (Pz) value was determined using the following equation below:(4)$$\mathrm{Pz}=\frac{\mathrm{Colony}\;\mathrm{diameter}}{(\mathrm{Precipitation}\;\mathrm{zone}+\mathrm{Colony}\;\mathrm{diameter})}$$

#### 4.6.2. Biofilm Formation 

The biofilm formation was evaluated by CLSM [76]. Briefly, one hundred microliters of yeast suspension (1 × 10^6^ cells) were seeded into a 96-well plate and incubated at 37 °C in a 5% CO_2_ humidified atmosphere for 4 h. Following the adhesion stage, the non-adherent yeast cells were removed and washed thrice with PBS. Then, 100 µL of niosomes were added and incubated continuously for 48 h. The medium was removed and the biofilm formation activity was determined by an MTT assay, as previously described. Next, the biofilms were stained with 10 µM of FUN-1 and 20 µg/mL of Con A-Alexa Flour 488 conjugate, incubated at 37 °C for 30 min, photographed, and analyzed by CLSM (Nikon AX; Nikon Instruments Inc., Melville, NY, USA). 

#### 4.6.3. Virulence-Related mRNA Expression 

To observe the genotypic expression levels of virulence factor-related genes, a quantitative reverse-transcription polymerase chain reaction (qRT-PCR) assay was performed. Niosome-treated *C. neoformans* was harvested and the total RNA was extracted using TRIZOL^®^ reagent (Invitrogen, Carlsbad, CA, USA) according to the manufacturer’s instructions. Total RNA was reverse transcribed into cDNA according to the RevertAid First Strand cDNA Synthesis Kit (Thermo Fisher Scientific, Waltham, MA, USA). The amplification was carried out by SYBR Green qPCR Master Mix (Thermo Fisher Scientific, MA, USA) and specific primers. The PCR primer sequences were designed according to *PLB1* (accession number CNAG_06085), *MAN1* (accession number CNAG_04312), *UGD1* (accession number CNAG_04969), *UXS1* (accession number CNAG_03322), and actin (*ACT1*) (accession number CNAG_00483). The sequences of the primers are listed in Table 3. The PCR reactions were performed in 35 cycles: initial denaturation at 94 °C for 30 s, annealing at 58 °C for 30 s, and extension at 70 °C for 60 s followed by cooling at 37 °C for 30 s. The mRNA expression levels were analyzed by the 2^−∆∆CT^ method and are expressed as the relative fold change when normalized with *ACT1* as a housekeeping gene [77].

### 4.7. Phagocytosis Assay 

The alveolar macrophages were activated by adding 0.6 μg/mL of LPS and 100 ng/mL of IFN-γ and incubated at 37 °C in a 5% CO_2_ humidified incubator for 24 h [79]. Nio-EEP (or Nio)-treated yeast was opsonized with a 1:10 dilution of anti-glucuronoxylomannan (GXM) monoclonal antibody (Clone 18b7) at 37 °C in a 5% CO_2_ humidified incubator for 1 h 30 min. The cells were then infected with 5 MOI of opsonized and niosome-treated yeast and incubated at 37 °C in a 5% CO_2_ humidified incubator for 2 h. After washing with PBS, the cells were subjected to Wright-Giemsa staining and the percentage of phagocytosis and phagocytosis index were determined using the following equations below [67]:(5)$$\%\mathrm{Phagocytosis}=\frac{\mathrm{Phagocytosed}\;\mathrm{cryptococci}}{\mathrm{Five-hundred}\;\mathrm{macrophages}}\;\times \;100$$
(6)$$Ph\mathrm{agocytosis}\;\mathrm{index}=\frac{\mathrm{Phagocytosed}\;\mathrm{cryptococci}}{\mathrm{Five-hundred}\;\mathrm{macrophages}}$$

Alternatively, after 2 h of the phagocytosis process, the supernatant was removed, washed, and continuously incubated at 37 °C in a 5% CO_2_ humidified incubator for 24 h in fresh medium. Subsequently, the cells were lysed using sterile deionized (DI) water for 30 min and then the intracellular yeasts were counted and expressed in CFU.

### 4.8. Statistical Analysis 

All data were presented as a mean ± standard error of the mean (SEM) in triplicate following three independent experiments. The Shapiro–Wilk test was used to check for a normal distribution and was followed by one-way analysis of variance (ANOVA) and Tukey’s post hoc test. Data without a normal distribution (% metabolic activity of A549 and NR8383) were analyzed using the Krustal–Wallis test and Dunn’s post hoc test. Significant differences (* *p* < 0.05) for all analyses were considered. All graphics were generated using Graph Pad Prism version 9.0 (GraphPad Software Inc., San Diego, CA, USA). 

## 5. Conclusions

In this study, Nio-EEP vesicles were successfully formulated on a nanometer scale with favorable physicochemical properties and were up-taken by *C. neoformans*. Moreover, the biological properties of Nio-EEP were introduced as anti-virulence factors, including the *PLB1* gene, biofilm formation, glycosyl components, and synthesis related genes, such as *UGD1* and *UXS1*. Furthermore, the intracellular replication of *C. neoformans* within alveolar macrophages was reduced after treatment with Nio-EEP. Regarding current studies, Nio-EEP could be a potential anti-virulence agent and be applied with multimodal treatments for pulmonary cryptococcosis.

## Figures and Tables

**Figure 1 molecules-28-06224-f001:**
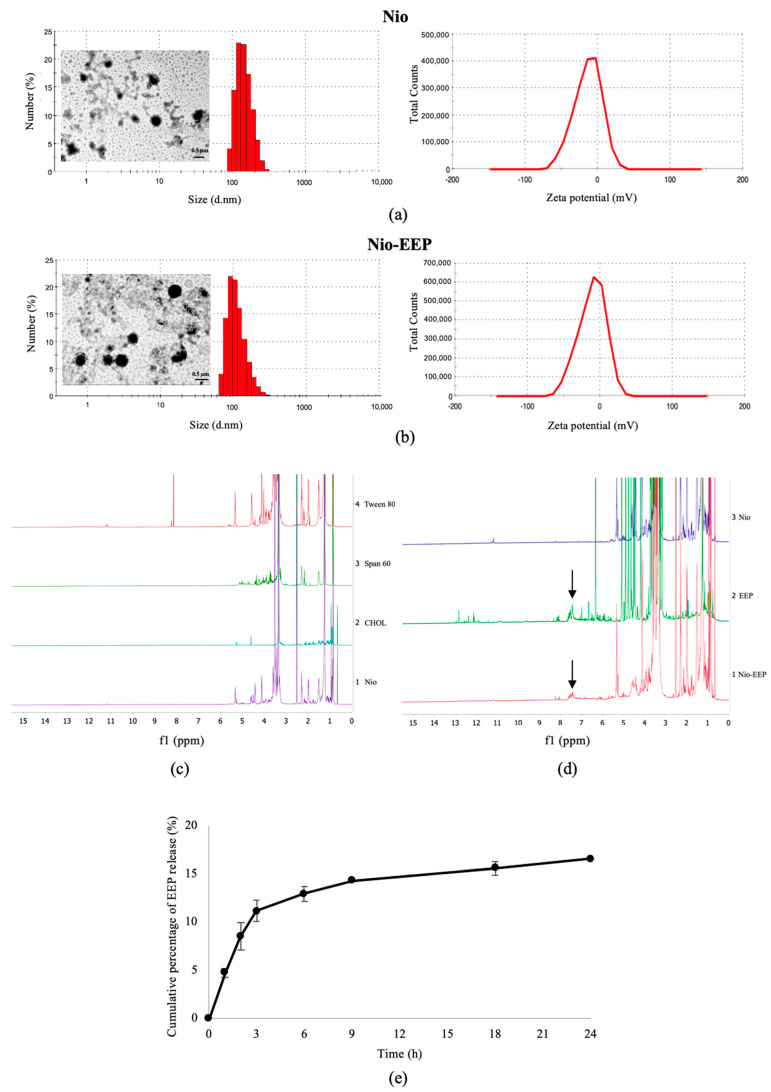
Characteristics of F1 formulation. Nanostructure, size distribution, and ZP distribution curves of (**a**) Nio and (**b**) Nio-EEP based on STEM and dynamic light scattering (DLS) analysis, respectively. The scale bar represents 0.5 µm. NMR spectra of (**c**) Nio composition and (**d**) Nio, EEP, and Nio-EEP are presented. (**e**) In vitro cumulative release of Nio-EEP in mSLF, pH 6.6, at 37 °C for 24 h. The data are represented as the mean ± SEM of three independent trials.

**Figure 2 molecules-28-06224-f002:**
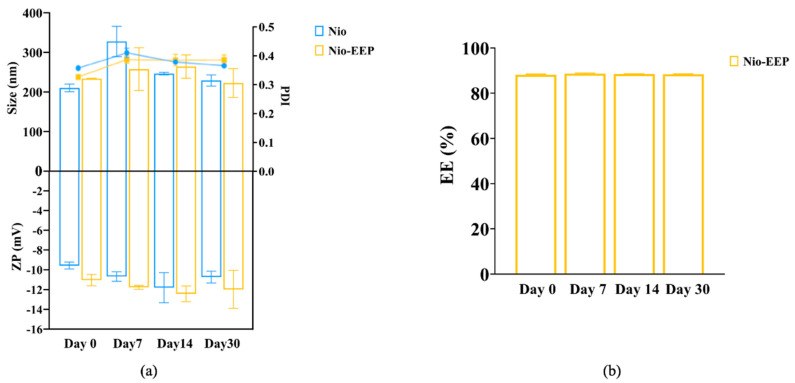
Stability of niosomes during storage at 4 °C for 1 month. (**a**) ● Blue dot and ■ Yellow square on the PDI line represent Nio and Nio-EEP, respectively. (**b**) Entrapment efficiency (EE) of Nio-EEP.

**Figure 3 molecules-28-06224-f003:**
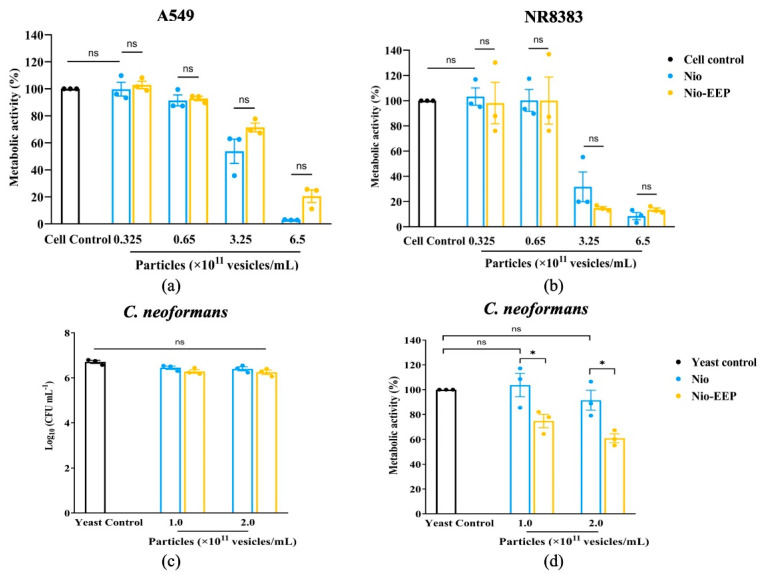
Determination of cytotoxicity and anti-fungal activity of Nio-EEP. Cytotoxicity of the niosomal formulations on (**a**) A549 and (**b**) NR8383 cell lines was evaluated by MTT assay. (**c**) Viability of treated yeast cells was assessed by CFU assay. (**d**) Reduction in metabolic activity in yeast cells from Nio-EEP. All values are expressed as mean ± SEM of three independent experiments performed in triplicate. Note: ns—not significant when compared to each group; * *p* < 0.05 when compared to each group. Black, blue, and yellow bars represent cell control or yeast control, Nio, and Nio-EEP, respectively.

**Figure 4 molecules-28-06224-f004:**
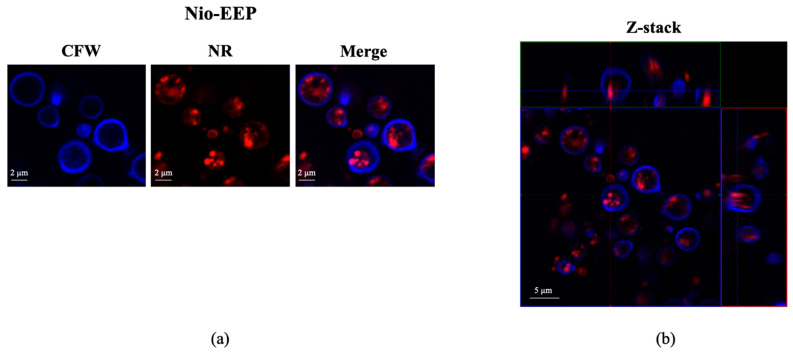
Intracellular uptake of niosomes by *C. neoformans*. (**a**) Localization of Nio-EEP inside the yeast cells. The Nio-EEP was pre-stained with NR (red) and subsequently incubated with CFW-labeled yeast cells (blue). The overlay of fluorescent images demonstrates the accumulation of Nio-EEP (red) within the yeast cells (blue). Scale bars represent 2 µm. (**b**) Orthogonal imaging analysis was performed to confirm the localization of Nio-EEP in the yeast cells. Scale bars represent 5 µm.

**Figure 5 molecules-28-06224-f005:**
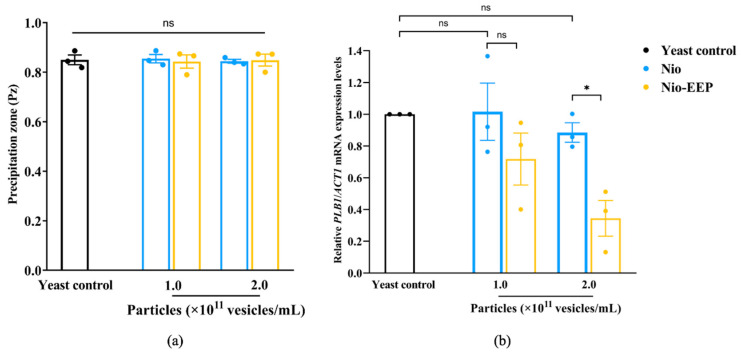
Effects of Nio-EEP on phospholipase production. (**a**) The production of phospholipase (Pz) was examined on EYA. (**b**) Down-regulation of *PLB1* on mRNA levels induced by Nio-EEP. Relative mRNA expression was normalized to *ACT1* and represented as a fold change in 2^−∆∆CT^ compared to the yeast control. Note: ns—not significant when compared to each group; * *p* < 0.05, significant compared to Nio. Black, blue, and yellow bars represent yeast control, Nio, and Nio-EEP, respectively.

**Figure 6 molecules-28-06224-f006:**
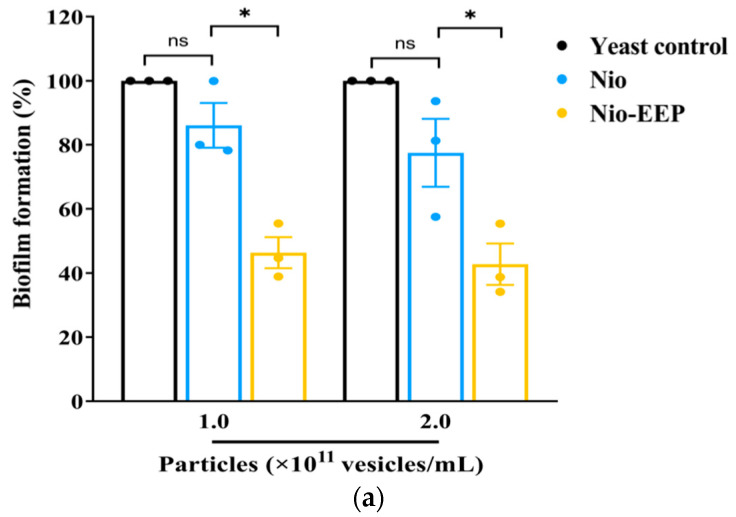
Reduction in biofilm formation by Nio-EEP. The surface-adhered yeast cells were treated with Nio-EEP for 48 h for mature biofilm. (**a**) Biofilm production was evaluated by an MTT assay. (**b**) Fluorescent and 3D images of biofilm thickness at 2 × 10^11^ vesicles/mL were taken by CLSM. The scale bar presents 50 µm. (**c**) Expression levels of biofilm-related genes, including *UGD1*, *UXS1*, and *MAN1* were assessed after Nio-EEP treatment. Relative mRNA expression was normalized to *ACT1* and expressed as a fold change. The error bars show mean ± SEM from three independent experiments performed in triplicate. Note: ns—not significant when compared to each group; * *p* < 0.05, significant compared to yeast control or Nio. Black, blue, and yellow bars represent yeast control, Nio, and Nio-EEP, respectively.

**Figure 7 molecules-28-06224-f007:**
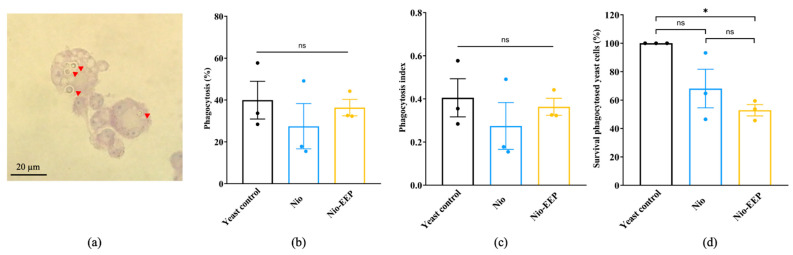
The Nio-EEP-induced killing of *C. neoformans* by alveolar macrophages. The treated yeast cells were opsonized with anti-GXM mAb (Clone 18b7) prior to infection in macrophages. (**a**) The macrophages infected with *C. neoformans* (red arrowhead) were stained by Wright-Giemsa, magnification 100×. (**b**) Phagocytosis (%). (**c**) Phagocytosis index determination. (**d**) Survival of treated yeast cells in the macrophages (%) after 24 h of incubation. The survival of yeast cells was assessed based on a CFU assay. The results show the mean ± SEM from three independent experiments performed in triplicate. Note: ns—not significant when compared to each group; * *p* < 0.05, significant compared to yeast control. Black, blue, and yellow bars represent yeast control, Nio, and Nio-EEP, respectively.

**Figure 8 molecules-28-06224-f008:**
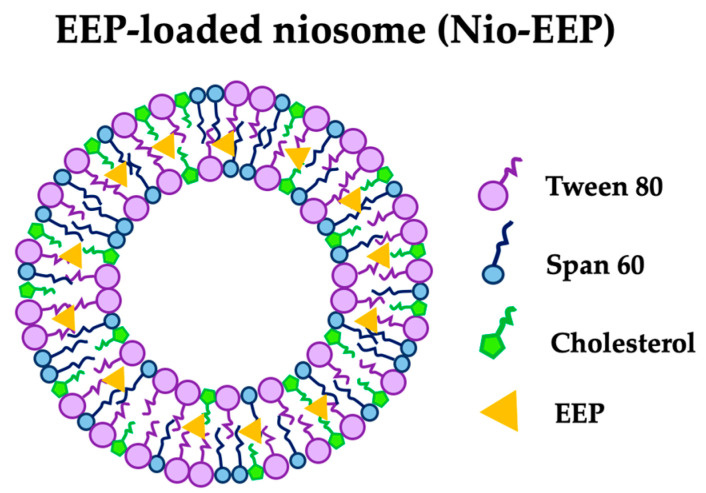
Schematic representation of a Nio-EEP.

**Table 1 molecules-28-06224-t001:** Particle size, polydispersity index (PDI), zeta potential (ZP), entrapment efficiency (EE), and loading capacity (LC) of the niosomes.

Formulations	Nio			Nio-EEP				
	Size (nm)	PDI	ZP (mV)	Size (nm)	PDI	ZP (mV)	EE (%)	LC (%)
F1	255.53 ± 25.36	0.37 ± 0.06	−9.38 ± 1.58	268.53 ± 10.89	0.32 ± 0.01	−10.54 ± 1.37	88.13 ± 0.01	82.95 * ± 0.01
F2	108.30 ± 6.53	0.34 ± 0.03	−8.62 ± 1.41	152.15 ± 34.00	0.35 ± 0.05	−10.05 ± 0.20	88.45 ± 0.00	45.76 ^#^ ± 0.00
F3	253.59 ± 20.49	0.32 ± 0.04	−10.51 ± 0.88	168.07 ± 23.91	0.32 ± 0.05	−9.92 ± 0.30	86.75 ± 0.05	50.49 ± 0.03

All data are represented as the mean ± SEM of three independent trials. * *p* < 0.05, significant for F1 compared to F2 and F3, and ^#^
*p* < 0.05, significant for F2 compared to F3.

**Table 2 molecules-28-06224-t002:** Composition of the Nio-EEP formulations.

Formulations	SP60: TW80: CHOL (mM Ratio)	SP60 (mg)	TW80 (mg)	CHOL (mg)	EEP (mg/mL)
F1	1:1:1	4.3	13.1	3.8	2
F2	2:2:1	8.6	26.2	3.8	2
F3	1:2:1	4.3	26.2	3.8	2

Abbreviations: SP60, Span 60; TW80, Tween 80; CHOL, Cholesterol; EEP, Ethanolic extract propolis.

**Table 3 molecules-28-06224-t003:** The specific primer sequences.

Primers	Primer Sequences (5′-3′)	References
*PLB1*	TGATGAATGAGAGCACGGAAGC	[78]
	CTCAGACCAGCCCAGTAGCT	
*MAN1*	GGCCTACGCTGAATTATGGA	This study
	GTAAAGAGCCGTCCTTGCAG	
*UGD1*	GAGGAGGCTTGTGCTAATGC	This study
	GACGACCTTGAAACCGATGT	
*UXS1*	AGCTGCATTTTACTCATCCCT	This study
	TCCTTGATGTAGGCGGGAGA	
*ACT1*	CCTTGCTCCTTCTTCTAT	[67]
	CTCGTCGTATTCGCTCTT	

## Data Availability

Not applicable.

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
