# Peer review of "Ethanolic Extract Propolis-Loaded Niosomes Diminish Phospholipase B1, Biofilm Formation, and Intracellular Replication of Cryptococcus neoformans in Macrophages"

_molecules, 2023, doi:10.3390/molecules28176224_

Round 1

Reviewer 1 Report

The authors prepared the formulation of ethanolic extract of propolis-loaded niosomes and evaluated the biological activities occurring during PLB1 production and biofilm formation of Cryptococcus neoformans. 

The work carried out is of contemporary interest. Moreover, niosomal preparation of a natural material has been used to achieve therapeutic effects in the lungs for the treatment of pulmonary cryptococcosis.

Manuscript is well written, experimental protocol is adequate. Results are aptly discussed and presented using graphics. Conclusion is drawn based on the obtained results.

Following are the general comments to further improve the quality of the MS:

1. Authors have used 1 HMR to study the chemical composition of surfactants, CHOL and Niosomes. At what MHz the signals were recorded? mention the make of 1HNMR instrument. However, the NMR spectrum does not indicate the exact chemical composition of propolis and conclusion drawn are quite vague as signals in the region of 7.25-7.5 are for aromatic protons. 

2. In line numbers 85, 124,138, 479 : in vitro sd be written in italic

3. In table 1, is data presented as mean plus minus SD or SEM. Mention it clearly in the legend of the table.

4. Expand MTT in line no 161 Mention the abbreviation at their first place of occurrence.

5. In line no165 and 249: "the results conclude that" sd be rephrased as Based on the results, it could be concluded that.... Because results do not conclude, we interpret based on the results. 

6. Line 570: do you mean Tukey's post hoc test (Written Turkey)?

Author Response

  1. Authors have used 1 HMR to study the chemical composition of surfactants, CHOL and Niosomes. At what MHz the signals were recorded? mention the make of 1HNMR instrument. However, the NMR spectrum does not indicate the exact chemical composition of propolis and conclusion drawn are quite vague as signals in the region of 7.25-7.5 are for aromatic protons. 

Answer: We have introduced the Nio-EEP Chemical Structure in the materials and methods section of the revised manuscript at page 16, line 535-542.

Thank you, the reviewer, for raising this valid point. The identification of the exact chemical composition of propolis using the NMR technique is quite challenging due to the complex nature of propolis. Indeed, NMR spectroscopy provides a piece of valuable information that sometimes cannot be achieved by the HPLC technique. The following text has been revised accordingly to clarify the chemical composition of propolis identified by 1H NMR analysis in the revised manuscript at page 3, line 120-132.

The 1H NMR spectrum of EEP revealed all phytochemical and other chemical constituents that can be interpreted based on chemical shift fingerprints. Examples of molecules with chemical shifts of aliphatic protons in the 0.5-3.0 ppm range include terpenoids, steroids, and linear fatty acid side chains for fats, oils, and waxes. Additionally, peaks in the chemical shift range of 3.5-5.5 ppm are due to sugar components. It is important to note that HPLC typically does not detect these compounds, thereby rendering NMR a useful alternative. Interestingly, chemical shifts around 6.0-8.1 ppm are also observed, which correspond to the protons belonging to aromatic of phenolic compounds [1, 2] . It was found that the 1H NMR spectrum of the Nio-EEP sample displayed chemical shifts that resembled the niosome components as well as the EEP components at a chemical shift between 7.25 and 7.75 ppm. In the HPLC analysis, our EEP sample consists of several phenolic compounds [3]. This result implies that phenolic compounds were successfully encapsulated into the niosomal formulation.

  1. In line numbers 85, 124, 138, 479 : in vitro sd be written in italic

Answer: We have already edited in the revised manuscript at line 88, 148, 383, 385, 570.

  1. In table 1, is data presented as mean plus minus SD or SEM. Mention it clearly in the legend of the table.

Answer: In the Table 1, we have clearly explained in the legend of the table that the data were mean±SEM and showed in the revised manuscript at page 4, line 143.

  1. Expand MTT in line no 161 Mention the abbreviation at their first place of occurrence.

Answer: We have already expanded the abbreviation of MTT in the revised manuscript at page 5, line 172.

  1. In line no165 and 249: "the results conclude that" sd be rephrased as Based on the results, it could be concluded that.... Because results do not conclude, we interpret based on the results. 

Answer: We have already edited in the revised manuscript at page 5, line 176 and page 10 line 290.

  1. Line 570: do you mean Tukey's post hoc test (Written Turkey)?

Answer:  We have already edited in the revised manuscript at page 19, line 657.

References:

  1. Kasote, D.M.; Pawar, M.V.; Bhatia, R.S.; Nandre, V.S.; Gundu, S.S.; Jagtap, S.D.; Kulkarni, M.V. “HPLC, NMR based chemical profiling and biological characterisation of indian propolis. Fitoterapia 2017, 122, 52-60, doi: 10.1016/j.fitote.2017.08.011.
  2. Tran, C.T.N.; Brooks, P.R.; Bryen, T.J.; Williams, S.; Berry, J.; Tavian, F.; McKee, B.; Tran, T.D. Quality assessment and chemical diversity of australian propolis from Apis mellifera bees. Sci Rep 2022, 12, 13574, doi: 10.1038/s41598-022-17955-w.
  3. Iadnut, A.; Mamoon, K.; Thammasit, P.; Pawichai, S.; Tima, S.; Preechasuth, K.; Kaewkod, T.; Tragoolpua, Y.; Tragoolpua, K. In vitro antifungal and antivirulence activities of biologically synthesized ethanolic extract of propolis-loaded PLGA nanoparticles against Candida albicans. Evid Based Complement Alternat Med 2019, 2019, 3715481, doi: 10.1155/2019/3715481.

Reviewer 2 Report

Dear authors 

Thank you for your effort in the manuscript entitled "Ethanolic Extract Propolis-Loaded Niosomes Diminish Phos pholipase B1, Biofilm Formation and Intracellular Replication of Cryptococcus neoformans in Macrophages". It is very interesting with significant results and study design. However please add the reason of choosing only F1, F2 and F3 with the chosen mM ratios at table 2. Moreover, some minor English mistakes were detected. 

Minor English editing is needed

Author Response

Q: However please add the reason of choosing only F1, F2 and F3 with the chosen mM ratios at table 2. Moreover, some minor English mistakes were detected. 

Answer:  We explained this point in the revised manuscript at page 11 line 333-339.

“ It was reported that these non-ionic surfactants were greater for high encapsulation of natural products [1, 2]and the EEP was successfully encapsulated by SP60 or TW80 with approximately 70% EE [3, 4] This study, the optimization of the different SP60 and TW80 concentrations was initiated according to Sangboonruang et al.[5]. Moreover, the ratio of surfactants was further optimized by reducing the concentrations of both SP60 and TW80 (F1), only SP60 (F3). In the case of reducing only TW80, we found the undesirable characteristic [5].”

References:

  1. Soliman, M.S.; Abd-Allah, F.I.; Hussain, T.; Saeed, N.M.; El-Sawy, H.S. Date seed oil loaded niosomes: Development, optimization and anti-inflammatory effect evaluation on rats. Drug Dev Ind Pharm 2018, 44, 1185-97, doi: 10.1080/03639045.2018.1438465.
  2. Chinembiri, T.N.; Gerber, M.; du Plessis, L.H.; du Preez, J.L.; Hamman, J.H.; du Plessis, J. Topical delivery of Withania somnifera crude extracts in niosomes and solid lipid nanoparticles. Pharmacogn Mag 2017, 13, 663-71, doi: 10.4103/pm.pm_489_16.
  3. Ilhan-Ayisigi, E.; Ulucan, F.; Saygili, E.; Saglam-Metiner, P.; Gulce-Iz, S.; Yesil-Celiktas, O. Nano-vesicular formulation of propolis and cytotoxic effects in a 3D spheroid model of lung cancer. J Sci Food Agric 2020, 100, 3525-35, doi: 10.1002/jsfa.10400.
  4. Patel, J.; Ketkar, S.; Patil, S.; Fearnley, J.; Mahadik, K.R.; Paradkar, A.R. Potentiating antimicrobial efficacy of propolis through niosomal-based system for administration. Integr Med Res 2015, 4, 94-101, doi: 10.1016/j.imr.2014.10.004
  5. Sangboonruang, S.; Semakul, N.; Obeid, M.A.; Ruano, M.; Kitidee, K.; Anukool, U.; Pringproa, K.; Chantawannakul, P.; Ferro, V.A.; Tragoolpua, Y.; et al. Potentiality of melittin-loaded niosomal vesicles against vancomycin-intermediate Staphylococcus aureus and staphylococcal skin infection. Int J Nanomedicine 2021, 16, 7639-61, doi: 10.2147/ijn.S325901.

Reviewer 3 Report

Dear editor and authors!

            The article entitled „Ethanolic Extract Propolis-Loaded Niosomes Diminish Phos3 pholipase B1, Biofilm Formation andIntracellular Replication of Cryptococcus neoformans in Macrophages” captures an interesting aspect of research on Cryptococcus and the action of natural compounds, on the example of propolis. The use of modern and safe carriers of drugs and compounds is becoming more and more common. The combination of these several aspects seems interesting.

I find the article interesting, well-crafted and appealing to the wider public of Molecules, however I recommend the following corrections (my specialty of expertise):

Major Revison

1. The introduction requires elaboration on biofilm in the context of Cryptococcus.

2.  As for the biofilm studies, they are interesting, but the CLSM results are very poorly presented. I recommend using Syto/PI staining. Photos should be larger, well presenting a given aspect, otherwise they make no sense, and the data should be better presented in the form of a graph. CLSM enables spatial imaging together with IMARIS. Maybe it's worth using. Which I personally suggest.

3. Fewer graph, or bigger and more readable. Maybe a table?

4. Complete the discussions with the latest developments in Cryptococcus biofilm research.

Author Response

Major Revision

  1. The introduction requires elaboration on biofilm in the context of 

As for the biofilm studies, they are interesting, but the CLSM results are very poorly presented. I recommend using Syto/PI staining. Photos should be larger, well presenting a given aspect, otherwise they make no sense, and the data should be better presented in the form of a graph. CLSM enables spatial imaging together with IMARIS. Maybe it's worth using. Which I personally suggest.

Answer:  We have already added the information of the biofilm in the revised manuscript at page 2, line 50-58.

According to the biofilm formation experiment, we thank for your suggestion for using SYTO9/PI staining. SYTO9 green fluorescent dye are permeate to bind to nucleic acid of both live and dead cells. Whereas, cells with a loss of plasma membrane integrity that are considered to be dead or dying will stain with propidium iodide (PI). Several studies used FUN1/ Concanavalin A conjugated to Alexa Fluor 488 (CAAF 488) [1], SYTO9 [2] or SYTO9/PI [3] for staining fungal biofilms. In this study, we focused on a cryptococcal biofilm and performed according to Bernaducci, et al. 2016 [1].  The FUN 1 exhibits orange-red fluorescent intravacuolar structures in metabolically active cells. While concanavalin A binds to mannoproteins in the matrix of cryptococcal biofilms. The visualization of the 3D structure of the biofilm community was photographed by CLSM (Nikon AX; Nikon Instruments Inc., NY, USA) and analyzed by NIS-Elements software. As your suggestions, we enlarged the size of CLSM images in Figure 6B in the revised manuscript. We also thank for your suggestion to use the IMARIS program.

  1. Fewer graph, or bigger and more readable. Maybe a table?

Answer: All images were corrected in details and size in the revised manuscript.

  1. Complete the discussions with the latest developments in Cryptococcus biofilm research.

Answer: We have already added the information of Cryptococcal biofilm research in the revised manuscript at page 13, line 444-448.

References:

  1.  Benaducci, T.; Sardi, J.; Lourencetti, N.; Scorzoni, L.; Gullo, L.F.; Rossi, S.; Derissi, J.; Prata, M.; Almeida, A.; Mendes, G.M.J. Virulence of Cryptococcus biofilms in vitro and in vivo using Galleria mellonella as an alternative model. Front Microbiol 2016, 7, 290, doi: 10.3389/fmicb.2016.00290.
  2.  Qian, W.; Li, X.; Liu, Q.; Lu, J.; Wang, T.; Zhang, Q. Antifungal and antibiofilm efficacy of paeonol treatment against biofilms comprising Candida albicans and/or Cryptococcus neoformans. Front Cell Infect Microbiol 2022, 12, 884793, doi: 3389/fcimb.2022.884793.
  3.  Shailaja, A.; Bruce, T.F.; Gerard, P.; Powell, R.R.; Pettigrew, C.A.; Kerrigan, J.L. Comparison of cell viability assessment and visualization of Aspergillus niger biofilm with two fluorescent probe staining methods. Biofilm 2022, 4, 100090, doi: 10.1016/j.bioflm.2022.100090.

Round 2

Reviewer 3 Report

Comments from the reviewer have been taken into account. The manuscript has been visually and substantively improved.